# Human Milk Oligosaccharide Profile across Lactation Stages in Israeli Women—A Prospective Observational Study

**DOI:** 10.3390/nu15112548

**Published:** 2023-05-30

**Authors:** Adi Talan Asher, Laurence Mangel, Julius Ben Ari, Ofer Gover, Wiessam Abu Ahmad, Jacky Herzlich, Dror Mandel, Betty Schwartz, Ronit Lubetzky

**Affiliations:** 1Institute of Biochemistry, Food Science and Nutrition, Robert H. Smith Faculty of Agriculture, Food and Environment, Hebrew University of Jerusalem, Rehovot 7610001, Israel; adi.talan@mail.huji.ac.il (A.T.A.); ofer.gover@mail.huji.ac.il (O.G.); 2Tel Aviv Medical Center, Department of Neonatology, Dana Dwek Children’s Hospital, Affiliated to the Sackler Faculty of Medicine, Tel Aviv University, Tel Aviv 6997801, Israel; laurencem@tlvmc.gov.il (L.M.); jackyh@tlvmc.gov.il (J.H.); ronitl@tlvmc.gov.il (R.L.); 3The Interdepartmental Equipment Facility, Robert H. Smith Faculty of Agriculture, Food and Environment, Hebrew University of Jerusalem, Rehovot 7610001, Israel; 4School of Public Health and Community Medicine, The Hebrew University of Jerusalem, Ein Kerem. P.O. Box 12271, Jerusalem 9112102, Israel; abua@savion.huji.ac.il

**Keywords:** human breast milk (HM), human milk indigestible oligosaccharides (HMOs), secretors, non-secretors, lactation stages

## Abstract

Human milk oligosaccharides (HMOs) stimulate the growth of gut commensals, prevent the adhesion of enteropathogens and modulate host immunity. The major factors influencing variations in the HMO profile are polymorphisms in the secretor (Se) or Lewis (Le) gene, which affect the activity of the enzymes fucoslytransferase 2 and 3 (FUT2 and FUT3) that lead to the formation of four major fucosylated and non-fucosylated oligosaccharides (OS). This pilot study aimed to determine the HMO profile of Israeli breastfeeding mothers of 16 term and 4 preterm infants, from a single tertiary center in the Tel Aviv area. Fifty-two human milk samples were collected from 20 mothers at three-time points: colostrum, transitional milk and mature milk. The concentrations of nine HMOs were assessed using liquid chromatography coupled with mass spectra chromatograms. Fifty-five percent of the mothers were secretors and 45% were non-secretors. Infant sex affected HMO levels depending on the maternal secretor status. Secretor mothers to boys had higher levels of FUT2-dependent OS and higher levels of disialyllacto-N-tetraose in the milk of mothers to girls, whereas non-secretor mothers to girls had higher levels of 3′-sialyllactose. In addition, the season at which the human milk samples were obtained affected the levels of some HMOs, resulting in significantly lower levels in the summer. Our findings provide novel information on the irregularity in the HMO profile among Israeli lactating women and identify several factors contributing to this variability.

## 1. Introduction

Human milk (HM) is the gold standard feeding regime for newborn infants and enhances infant health, growth, and development [1]. HM is a dynamic biological fluid which is highly variable among individuals. Its composition varies depending on the infant’s gestational stage (preterm versus term) and as lactation progresses to meet the needs of the growing infant [2]. One of the most important bioactive components in HM are the HMOs, a highly abundant group of indigestible oligosaccharides that stimulate the growth of gut commensals, prevent the adhesion of enteropathogens and modulate host immunity [3]. Human breast milk contains 12~13 g/L of HMOs, of which more than 250 varieties have been separated and more than 200 chemical structures have been characterized to date [4,5,6]. Of 200 characterized HMOs, around 170 structures are listed in the tables [7,8]. The concentrations of the representing HMOs have been determined in secretor and non-secretor donor’s milk at several lactation periods [9]. The average values of the concentrations of 2′-fucosyllactose (2′-FL), 3-fucosyllactose (3-FL), lacto-N-tetraose (LNT), 3′-sialyllactose (3′-SL) and 6′-sialyllactose (6′-SL) have been calculated from the published data by means of weighted analysis [10]. The HMOs are synthesized in the mammary gland and are classified into three major types depending on the expression and activity of the enzymes sialyltransferases and fucosyltransferases (FUT), resulting in neutral fucosylated HMO (e.g., 2′-FL), neutral N-containing HMOs (e.g., LNT) and acidic (sialylated) HMOs (e.g., 6′-SL, disialyllacto-N-tetraose: DSLNT) [11]. The mother’s genetic makeup determines the activity of the α1-2-fucosyltransferase (FUT2) encoded by the secretor (Se) gene and the α1-3/4-fucosyltransferase (FUT3) encoded by the Lewis (Le) genes, resulting in four milk phenotypes (secretors: *Se+Le+*, non-secretors: *Se+Le−*, *Se−Le+*, and *Se−Le−*) [12,13,14,15]. The four major milk groups are determined by the presence or absence of fucosylated oligosaccharides (OS) in HM and are regulated by the activity of two fucosyltransferases (FUT) enzymes; the Se gene encodes FUT2 for the generation of α1,2-fucosylated OS, and the FUT3 enzyme is encoded by the Le gene, for the generation of α1, α1,3/4-fucosylated OS, and based on the Lewis blood group system. Women with an active Se locus are classified as secretors (*Se+*), whereas women with an active Le locus are classified as Lewis-positive (Le+). Women lacking FUT2 or FUT3 activity are classified as non-secretors (*Se−*) or Lewis negative (*Le−*), missing α1,2-fucosylated or α1, 3/4-fucosylated OS, respectively [12,13,14,15]. Non-genetic factors such as gestational age (GA) at delivery, stage of lactation, seasons of lactation, sociodemographic and environmental factors have been reported to play a role in the composition and concentration of HMOs [13,16,17,18,19,20].

HMO levels have been determined in several regions of the world, such as in China [21], Mexico [22], Vietnam [23] and Samoa [24]. Since little is known about the HMO content and variation in HM in our region, we sought to conduct this longitudinal analysis to assess the HMO profile across lactation stages in lactating Israeli women.

## 2. Materials and Methods

### 2.1. Study Design and Participants

This prospective cohort pilot study was conducted in accordance with the Declaration of Helsinki, and approved by the Institutional Ethics Committee of the Medical Research, Infrastructure, and Health Services Fund of the Tel Aviv Medical Center. The project’s identification name was “The Role of Bioactive Molecules Present in Preterm Human Breast Milk in the Inhibition of Experimental Necrotizing Enterocolitis”; Code: 0001-20-TLV; Date of approval: 13 May 2020. Written informed consent was obtained from all participants before study entry. Healthy lactating mothers of term and preterm infants were recruited at the Neonatology Department of the Dana-Dwek Children’s Hospital of the Tel Aviv Medical Center between August 2020 and August 2021. The collected demographic and clinical data included maternal age, BMI, diet and habits, mode of delivery, parity, infant gestational age, gender, birth weight, and admission to NICU. Seasons of HM sampling were recorded. The collected samples were assigned into two seasons, summer, from June to August, and not summer, from September to May.

### 2.2. Human Milk Samples

HM samples from healthy mothers were collected at different time points: colostrum ≤ 72 h, transitional milk (between 5 and 15 days) and mature milk (between 31 and 52 days) at mother’s convenience, either by manual expression or using a double electric pump. Samples were stored frozen until shipment via cold chain transportation for laboratory analyses. The maternal status of “secretor” (*Se+*) was based upon the presence/absence of 2′-FL HMO, which indicates the activity of FUT2. Maternal Lewis blood type (*Le+*) was based upon the presence of lacto-N-fucopentaose II (LNFP II), which indicates the activity of FUT3 [25,26].

#### 2.2.1. Chemical Analysis and OS Composition

We selected nine major HMOs for the analysis, which included 2′-FL (Cat #: L302, Dextra laboratories, Reading RG2 9LH; UK), 3-FL (Cat #: L303, Dextra laboratories, UK), lacto-N-fucopentaose I (LNFP I, Cat #: L502, Dextra laboratories, Reading RG2 9LH; UK), LNFP II (Cat #: L503, Dextra laboratories, UK), lacto-N-fucopentaose III (LNFP III, Cat #: L504, Dextra laboratories, Reading RG2 9LH; UK), LNT (Cat #: L403, Dextra laboratories, Reading RG2 9LH; UK), 3′-SL (Cat #: SL302, Dextra laboratories, Reading RG2 9LH; UK), 6′-SL (Cat #: SL306, Dextra laboratories, Reading RG2 9LH; UK) and disialyllacto-N-tetraose (DSLNT, Cat #: SLN603, Dextra laboratories, Reading RG2 9LH; UK).

HM samples were thawed and pasteurized in a water bath at 62.5 °C for 30 min. Samples were mixed with an equal volume (1:1) of double distilled water and centrifuged at 4 °C for 30 min at 4000× *g* to remove the protein and the lipid layer. The intermediate layer of milk protein and free oligosaccharides were collected. The extract was then mixed with two volumes of ethanol (1:2) and placed at 4 °C for 4 h, centrifuged at 10,000 rpm for 30 min at 4 °C, and the upper free OS layer was collected. Filtered OS samples were lyophilized and stored at −80 °C [27] until analysis. Calibration curves were made using HMO standards. Results were calculated and provided in concentration units of nmol/mL.

#### 2.2.2. LC-MS Analysis

OS samples were analyzed using the LC-MS system, which consisted of Dionex Ultimate 3000 RS HPLC coupled to a Q Exactive Plus hybrid FT mass spectrometer equipped with a heated electrospray ionization source (Thermo Fisher Scientific Inc.; Paisley PA4 9RF, UK). The analytes were separated on an HILIC-OH5 column (2.1 × 150 mm, 2.7 µm, Agilent) employing linear gradient of acetonitrile (solvent B)/water with 20 mM ammonium formate pH 4.3 (solvent A). The mass spectrometer was operated in negative ESI full scan and PRM (targeted MS/MS) modes. Ion source parameters were as follows: spray voltage 3.7 kV, capillary temperature 250 °C, sheath gas rate (arb) 40, and aux. and sweep gas rates (arb) 10, aux. gas temp. 250 °C. Mass spectra were acquired in the m/z 200–1500 Da range at resolving powers of 70,000 (scan mode) and 35,000 (PRM). The LC-MS system was controlled using Xcalibur software (Xcalibur 4.3, Thermo Fisher Scientific Inc., Waltham, MA, USA).

By using the HILIC-OH5 column, some HMOs were not separated chromatographically (see Appendix A, and Appendix A). However, these compounds have distinctive MS/MS spectra and can be differentiated by different product ions (see Appendix A for MS/MS spectra of individual HMOs). The nine HMOs were determined and quantified from the calibration curves prepared in ethanol/water (1:1) from commercially available HMOs standards at the following concentrations, 5, 12.5, 50, 125, 250, 500, 1250, 2500, 5000 ng/mL. The lyophilized samples were dissolved before the analysis in 1 mL of the same solvent. Calibration samples were run separately for each batch of samples because calibration was performed in external mode due to the absence of appropriate internal standards. Data were analyzed using TraceFinder software (TraceFinde 5.1, Thermo Fisher Scientific Inc., Waltham, MA, USA).

#### 2.2.3. GC-MS Analysis

The OS samples were further analyzed using GC-MS, a gas chromatograph (Agilent 7890A) coupled with the mass selective (Agilent 5975C MSD) detector. The compounds were separated on Rxi5-MS capillary column (30 m × 0.25 mm, 0.25 µm, Restek). Helium was used as a carrier gas at a 1.3 mL/min flow rate. Before the analysis, the samples were derivatized with trimethylsilylation reagent, which consisted of pyridine, BSA, and TMCS (20:20:1). Analytical equipment was controlled, and data were analyzed using MassHunter Acquisition and Data Analysis software (MassHunter 12.0, Agilent, Santa Clara, CA, USA). The GC-MS was used for monitoring the purified fractions for the presence of mono- and disaccharides. GC-MS cannot be and thus was not used to quantitative determine HMOs, even after chemical derivatization, due to their high molecular weights.

### 2.3. Statistical Analysis

Statistical analysis was performed using the Stata/SE version 15.0 (StataCorp) and R statistical software version 3.5.0 (R Project for Statistical Computing) unless otherwise stated. Descriptive statistics are given as mean ± SEM or as frequency (n) with percentage (%), according to the scale of the variable. Linear and logistic regression mixed-effect models with a random intercept at the mother level were applied to assess differences between measurements and correlations between baseline characteristics, birth outcomes, and pregnancy outcomes. To assess differences in time points by Secretor status (Se+ or Se−) and in infants’ sex by Secretor status, each HMO was modeled by an interaction term, Secretor status*time of lactation or Secretor status*infant sex, respectively, followed by a series of pairwise comparisons. All significant variables at an alpha level of 0.2 were entered into multivariable regression models, taking into account clinical considerations, to avoid unstable models due to the small sample size limitation. ANOVA and Tukey HSD by JMP pro 15 version (SAS Institute Inc., Cary, NC, USA, 1989–2019) were conducted to compare between levels of HMOs. A *p*-value < 0.05 was considered statistically significant. All reported *p*-values are two-tailed.

All graphs were made using GraphPad Prism version 8.0.0 for Windows, GraphPad Software, San Diego, CA, USA, www.graphpad.com, accessed on 15 February, 2023

## 3. Results

A total of 21 mothers agreed to participate in the study, after the exclusion of one mother due to her use of probiotics, 20 mothers were included in the study, including 16 mothers of term infants and 4 mothers of preterm infants (GA < 37 weeks). They contributed 55 human milk samples, but two samples were excluded due to insufficient volume for chemical analysis. The remaining 53 samples consisted of 19 colostrum, 18 transition and 16 mature milk samples. Table 1 depicts the maternal and neonatal characteristics of the cohort.

The proportion of secretor mothers was 55% (11/20), and that of non-secretors was 45% (9/20). The proportion of Lewis-positive mothers was 85% (17/20), and Lewis-negative 15% (3/20). Therefore, 50% (10/20) of the mothers were considered to have the *Se+Le+* phenotype, 35% (7/20) had the *Se−Le+* phenotype, 5% (1/20) had the *Se+Le−* phenotype and 10% (2/20) had the *Se−Le−* phenotype.

Their HMO profiles categorized by secretor status and lactation stage are shown in Figure 1. In secretors, 2′-FL was the dominant OS throughout the course of lactation (Figure 1A–C). In non-secretors, LNT was predominant in colostrum (Figure 1D) and transitional milk (Figure 1E), while 3-FL was the most abundant OS in mature HM samples (Figure 1F). Appendix A provides a summary of the mean values for individual HMO concentrations of secretor, non-secretor and all HM samples.

Throughout lactation, the HMO profile in secretors and non-secretors displayed significant differences (Figure 2 and Table 2). FUT2-dependent HMOs, 2′-FL and LNFP I, were only detected in HM samples from secretors. In multivariate analysis, 2′-FL levels significantly decreased as lactation progressed towards mature milk (*p* < 0.001) and LNFP I levels did not change significantly. 3-FL was significantly enriched in non-secretor colostrum and transition milk samples compared with secretors (*p* < 0.001); however, 3-FL levels remained stable throughout lactation in both secretor and non-secretor HM phenotypes. LNT levels in colostrum were significantly higher in non-secretors than in secretors (*p* < 0.001). LNT levels decreased significantly from transitional to mature milk in the non-secretors (*p* < 0.01), whereas in secretors, the LNT levels increased over the course of lactation (*p* < 0.01). 3′-SL levels were significantly higher in colostrum samples of secretors than non-secretors (*p* < 0.001) and decreased as lactation progressed in both secretors and non-secretors (*p* < 0.001). In non-secretors, 6′-SL levels significantly decreased during the course of lactation (*p* < 0.01), whereas in secretors, variations in 6′-SL levels were not significant. Colostrum content of DSLNT and LNFP III were higher in non-secretors than in secretors (*p* < 0.05 and *p* < 0.001, respectively), and their levels decreased significantly throughout lactation (*p* < 0.001 and *p* < 0.01, respectively). Finally, LNFP II levels remained stable through all time points in secretors, and in non-secretors, levels were significantly lower in mature milk than in transitional milk samples *p* < 0.05).

Furthermore, the season at which HM samples were obtained exerted a notable impact on the LNT, LNFP II, and LNFP III levels, with significantly lower levels in the summer (June–August) (*p* < 0.05, *p* < 0.01 and *p* < 0.001, respectively) (Figure 3 and Table 2).

Infant sex also affected HMOs levels. Regardless of secretor status, 2′-FL and LNFP I concentrations were significantly higher in mothers to boys (*p* < 0.001 and *p* < 0.05, respectively), while 3′-SL and DSLNT were significantly higher in mothers to girls (*p* < 0.01 and *p* < 0.05, respectively) (Figure 4 and Table 2). When considering secretor status, compared with mothers to boys, 3′-SL concentrations were higher in secretor mothers to girls (*p* < 0.001) and DSLNT concentrations were higher in non-secretor mothers to girls (*p* < 0.001). Additionally, significant 2′-FL concentrations were found in non-secretor mothers to boys (Figure 4).

## 4. Discussion

In this study, we showed that 55% of the Israeli women were secretors. These numbers are lower than those reported for Brazilian (81.2%) [15], Bangladesh (66%) [28], Germany (85.3%) [29], and other European countries (77%) [18]. A recent cross-sectional study comprising 410 healthy breastfeeding mothers from 11 international cohorts reported 72% of secretor mothers [16]. These specific inherent genetic population variations may likely reflect natural evolution selection driving reduced α1–2-fucosylated HMOs, which may ultimately influence the infant’s disease risk or maturation. Additionally, sociodemographic factors may contribute to the human milk HMO secretor composition [30].

McGuire et al. [31] had previously established that HMO concentrations vary due to geographic location. Our study contributes new information about HMO composition in healthy Israeli mothers. When comparing all of our samples to the 11-country cohort reported by McGuire et al. [31], our results suggested that the 2’FL concentration (1012 ± 175.6 nmol/mL) of Israeli women is the lowest documented. It is closer to those collected in Ghana (1438 + 207 nmol/mL), but six to seven times lower than in Peru (6528 + 435 nmol/mL) and in the United States California (USC) (7043 ± 858 nmol/mL). In milk produced by Israeli women, the concentration of LNT (474.1 ± 60.54 nmol/mL) and DSLNT (172.4 ± 11.79 nmol/mL) also appeared to be lower than that reported in the cohort (Peru LNT 953 + 139 nmol/mL and in Sweden DSLNT 216 + 14). This trend is kept also for other sialylated HMOs (3′SL and 6′SL) in Israeli HM. However, 3 FL concentration is found to be within the concentration range of the cohort, with great similarity to those in Peru, Spain, Sweden, USC, and in rural areas in Ethiopia. Furthermore, in the cohort, LNFP III was significantly higher in Sweden (269 + 22 nmol/mL) than in all other cohorts (*p* < 0.05) except for USC (76 ± 10 nmol/mL) and perhaps except for Israel (103.8 ± 17.42 nmol/mL). The reasons for these ethnic differences are currently unknown and should be further investigated [31].

We found expected differences in HMO concentrations between secretors and non-secretors [32]. Secretors’ milk contains large amounts of α1-2 fucosylated oligosaccharides such as 2′-FL and LNFP I, explained by the activity of FUT2. We found that 2′-FL was the predominant HMO in the secretors’ milk in all three collection periods, and that LNT was the major HMO in colostrum and transitional milk samples in non-secretors’ milk, similar to the findings reported for the Brazilian cohort [15]. In non-secretors’ milk, LNT, 3-FL, and LNFP II levels were in good agreement with the results recently published by Liu et al. [33]. After one month of lactation, we found that 3-FL was the most abundant HMO in non-secretors’ milk. We also found that 3-FL levels were positively correlated with the secretor status. Significantly higher concentrations of 3-FL in non-secretors’ milk were found during lactation stages at the first month of lactation compared to the secretors’ milk. These results are consistent with Samuel et al. [18] and Thurl et al. [34], who found that 3-FL is predominantly biosynthesized in the milk of non-secretors, especially of milk group 2 (*Se−Le+*), together with LNFP II. These two major HMOs, 3-FL and LNFP II, may protect newborns against infections by blocking a fucose-binding lectin of the human pathogen *Pseudomonas aeruginosa* [35]. Lectins are frequently proposed as potential immune cell receptors for circulating HMOs [14]. LNT binds to galectin-8, a key receptor involved in defense against intracellular bacterial pathogens [36]. Both 2′-FL and 3-FL bind the dendritic cell-specific cell surface receptor DC-SIGN with a high degree of specificity, potentially modulating the behavior of these key drivers of adaptive immune cell maturation [37]. Moreover, 3-FL was found to affect colon motor contractions ex vivo and was demonstrated to diminish colon motor contraction twice as much as 2′-FL [38], therefore possibly compensating for the lack of 2′-FL in non-secretors and the overall significantly lower concentration of 3-FL relative to 2′-FL. Both 3-FL and 2′-FL may play a role in the infant’s gastrointestinal motor function, which develops after birth [39]. During various lactation stages, non-secretors’ milk was characterized by higher concentrations of LNT [30,32], LNFP II [25,32,34], and LNFP III [32]. Although we did not find these HMOs to be associated with secretor status, they were significantly higher in non-secretors’ milk when considering the time of lactation. LNT and LNFP III were enriched in non-secretors’ colostrum, while LNFP II was enriched in their transitional milk. LNT is a neutral nonfucosylated HMO core structure for many fucosylated HMOs (e.g., LNFP-I, -II). When FUT2 is inactive, other enzymes can access this limited availability core structure [18] and thus create an HMO profile that might compensate for the absence of α1,2-fucosylated HMO’s in non-secretors milk.

We found significantly higher levels of DSLNT in non-secretor colostrum, which is a non-fucosylated but sialylated LNT-based HMO, and these levels are directly correlated with the levels of LNT. This finding is consistent with the findings reported for healthy European mothers [18]. DSLNT is an HMO that plays a crucial role in protecting the infant from the devastating inflammatory disease NEC [40,41]. This was shown in in vivo experiments with a neonatal rat model [42]. The putative protective effect exerted by DSLNT is a highly associated structure-specific effect [43]; thus, infants of non-secretors may be protected by high concentrations of DSLNT, while infants of secretors further benefit from the protection of 2′-FL. 2′-FL was found to inhibit toll-like receptor 4 (TLR4) signaling by directly binding to TLR4 [44], which is known to be activated by bacterial endotoxins known to play a key role in initiating the innate immune response responsible for chronic and acute inflammatory disorders [45,46]. Throughout lactation, this difference in DSLNT concentrations between the two phenotypes was found to be significant only in the colostrum, with a significant decrease in both DSLNT and 2′-FL throughout lactation. Therefore, this compensation may be due to the content of immune system-associated molecules in the colostrum and may be related to the protection needed in the infant’s first days of life. However, in our model, we did not find any association between DSLNT and the secretor status, which was demonstrated in the study of Azad et al. [16].

We found that 3′-SL was associated with secretor status, with a higher concentration in secretors’ milk; however, comparing its concentration between the two groups demonstrated a significant difference only in the colostrum. Our findings are in good agreement with the recently published finding of Selma-Royo et al. [47], but contrast the studies reported by Samuel et al. [18] and Azad et al. [16]. 3′-SL and 6′-SL were found to be less affected by milk groups but affected by the time of lactation. We did find for both groups a depletion of 3′-SL levels throughout the first month of lactation. Some published studies show that concentrations of acidic HMOs decrease significantly during the first three months postpartum [9]. In addition to the decrease in DSLNT and 3′-SL, we found a significant decrease in non-secretors’ 6′-SL. However, secretor 6′-SL did not vary during lactation stages, along with other HMOs concentrations that remained stable. These results are in contrast with previous studies that have demonstrated a significant decline in secretors HMOs in the first month of lactation [48]. Previous research showed that most HMO content other than 3-FL declines throughout lactation [34], which agrees with our present results when not considering the secretor status. While the colostrum contains more immunoglobins and other nutrients that help with building the baby’s immune system, transitional milk stands for a period of increased milk production to support the nutritional and developmental requirements of the rapidly growing infant [49]. We found a significant peak in concentrations of LNT, LNFP II and 6′-SL in transitional milk. In a pilot study including 49 mother–infant pairs, higher breastmilk concentrations of LNFP II at two weeks were associated with a lower risk of respiratory and gastrointestinal illnesses at 6 and 12 weeks in infants [50]. 6′-SL is known to support infant brain development by supplying sialic acid, an essential building block for neurons [51]. Thus, feeding with 3′-SL and 6′-SL increased the level of gangliosides in cerebellum and corpus callosum in an in vivo experiment with piglets [52]. These are crucial for successful feeding and growth; thus, in this sense, increasing HMOs levels moving from the colostrum to transitional milk may be due to developmental importance.

We found that during summer, HBM samples had significantly lower concentrations of LNT, LNFP II, and LNFP III compared to the other seasons. Azad et al. [16] found that LNFP III concentrations were significantly lower in milk collected during winter or spring compared with summer in the Canadian population. A study of rural African Gambian women [53] found that HMO concentrations were significantly higher in the dry season (when food is more plentiful, which can lead to higher energy intake), although individual HMOs were not examined. In the case of our study population of Israeli women living in Tel Aviv, other seasonal factors may contribute to these observed changes in HMO levels rather than energy intake, and thus need further examination.

We found a significant variation in HMO levels associated with infants’ sex. We found that 2′-FL and LNFP I concentrations, FUT2-dependent HMs, were significantly higher in the milk of mothers to boys. This finding is supported by the results of Zimmermann and Curtis, who reviewed 44 studies investigating 3105 breast milk samples from 2655 women [54]. In contrast, Azad et al. and Masi et al. reported that infant sex was not associated with HMO composition [16,41]. Our finding is also in good agreement with those of Wang et al. [30], who showed that in secretors, mothers of boys had higher concentrations of LNFP-I. However, Tonon et al. [55] found higher 2′-FL concentration in the milk of mothers of girls [55]. Other researchers found a positive correlation between 2′-FL concentrations in HBM from secretors and infant growth [56] or with weight gain during 0 to 5 months [57]. The significantly high concentration of 3-FL that was found in non-secretor mothers of boys can be explained by a matter of possible compensation for the lack of 2′-FL in non-secretors, as previously described. We also found significantly higher concentrations of 3′-SL and DSLNT in the breast milk of mothers to girls rather than in boys. This trend was significant for secretors 3′-SL and for non-secretors DSLNT, which can also be explained by possible compensation between these two phenotypes. In a Gambian study, the sialylatet OS 3′-SL was positively correlated with weight development [58], while DSLNT has been demonstrated to be positively associated with fat mass in mature milk one-month postnatal infants [59]. The differences in DSLNT content in term HBM of ≥1 month may be driven partly by biological differences in body composition between the sexes. Body composition differences are known to emerge during the fetal and postnatal periods. Following birth, females generally have greater fat mass and less fat-free mass [30].

A limitation of our study is that participants were recruited in a single tertiary center with a rather homogenous socioeconomic background from central Israel. In addition, some analyses did not show significance despite seemingly apparent differences due to the small size of the cohort.

This study is the first report on the variability of the HMO profile in lactating Israeli women from the Tel Aviv area. We demonstrated that the HMO profile depended upon maternal (secretor status) and infant factors (infant sex) as well as environmental factors (season) and varied as lactation progressed. Future studies are warranted to confirm our findings in larger cohorts that include diverse Israeli communities.

## Figures and Tables

**Figure 1 nutrients-15-02548-f001:**
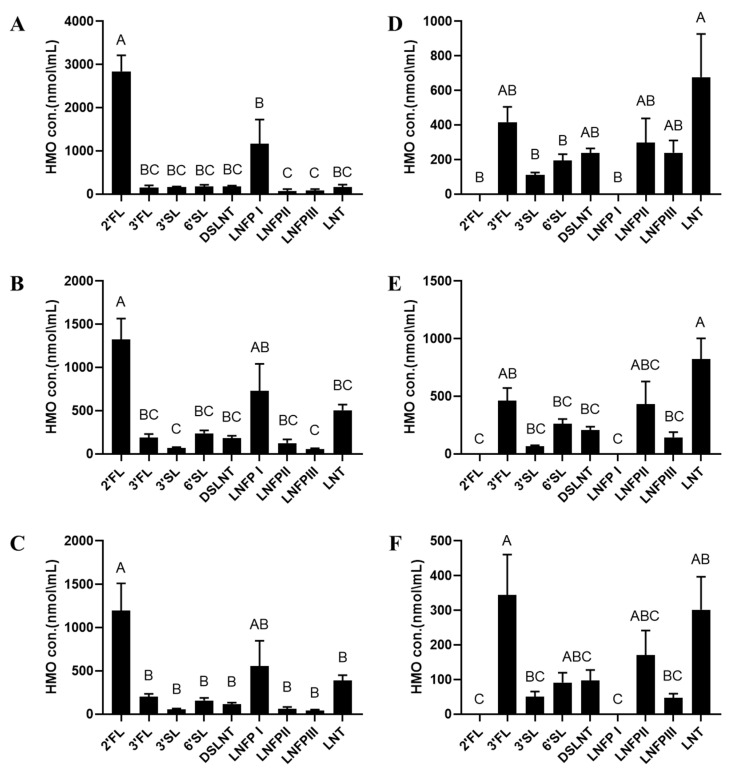
HMO profile of secretors and non-secretors at different lactation stages. Left panel, secretors, right panel, non-secretors. (**A**,**D**) Colostrum < 72 h: secretors, *n* = 10; non- secretors, *n* = 9. (**B**,**E**) Transitional milk ≤ 15 days: secretors, *n* = 10; non-secretors, *n* = 8. (**C**,**F**) Mature milk ≥ 1 month: secretors, *n* = 10; non-secretors, *n* = 6. Statistics according to Tukey’s multiple comparisons test. Data are expressed as mean ± standard error of mean (SEM). Different letters indicate statistical significance.

**Figure 2 nutrients-15-02548-f002:**
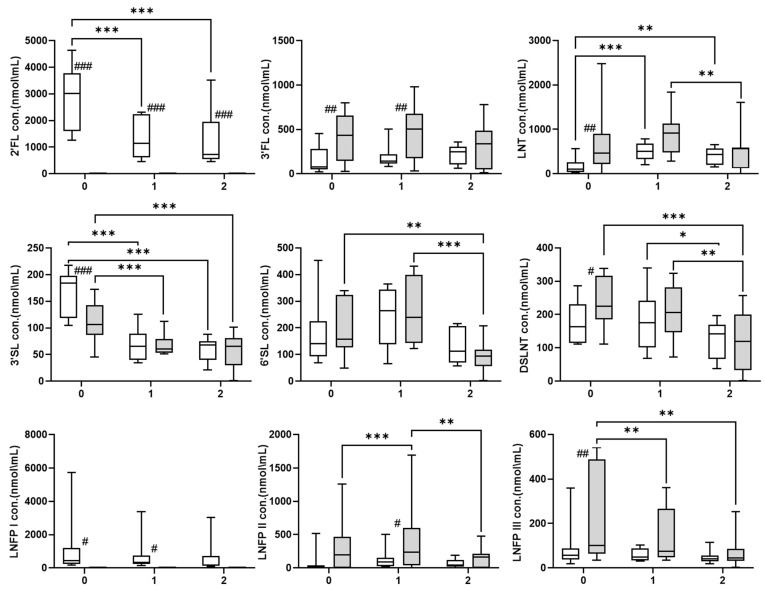
Box plot showing concentrations (mean ± SEM) of the individual HMOs by secretor status and lactation stages. (0) Colostrum < 72 h: secretors, *n* = 10; non-secretors, *n* = 9. (1) Transitional milk ≤ 15 days: secretors, *n* = 10; non-secretors, *n* = 8. (2) Mature milk ≥ 1 month: secretors, *n* = 10; non- secretors, *n* = 6. * *p* < 0.05, ** *p* < 0.01, *** *p* < 0.001 indicate significant differences within the group (secretors or non-secretors) between lactation stages. # *p* < 0.05, ## *p* < 0.01, and ### *p* < 0.001 indicate significant differences between the groups (secretors versus non-secretors) at each lactation stage. Multivariate analysis was used for comparisons, additional detailed results are presented in Table 2. SEM = standard error of mean.

**Figure 3 nutrients-15-02548-f003:**
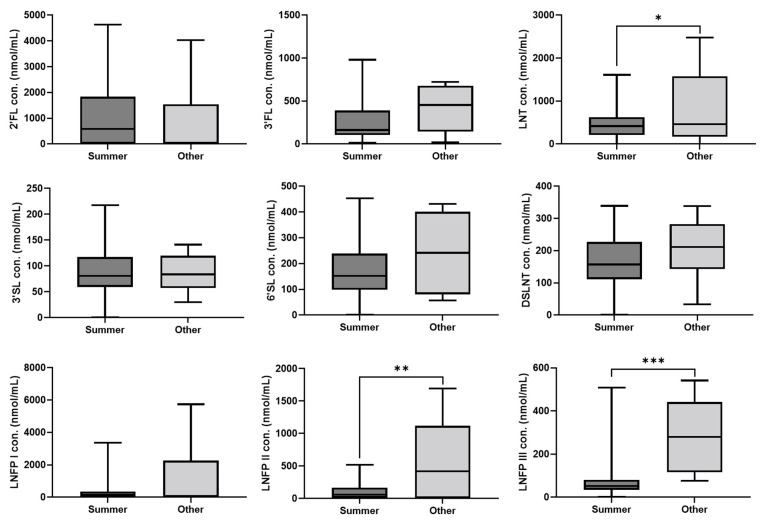
Box plot showing concentrations (mean ± SEM) of the individual HMOs by season. Summer: June–August, *n* = 45; Other: September–May, *n* = 8. * *p* < 0.05, ** *p* < 0.01, *** *p* < 0.001; Multivariant analysis was used for comparisons, and detailed results are presented in Table 2. SEM = standard error of mean.

**Figure 4 nutrients-15-02548-f004:**
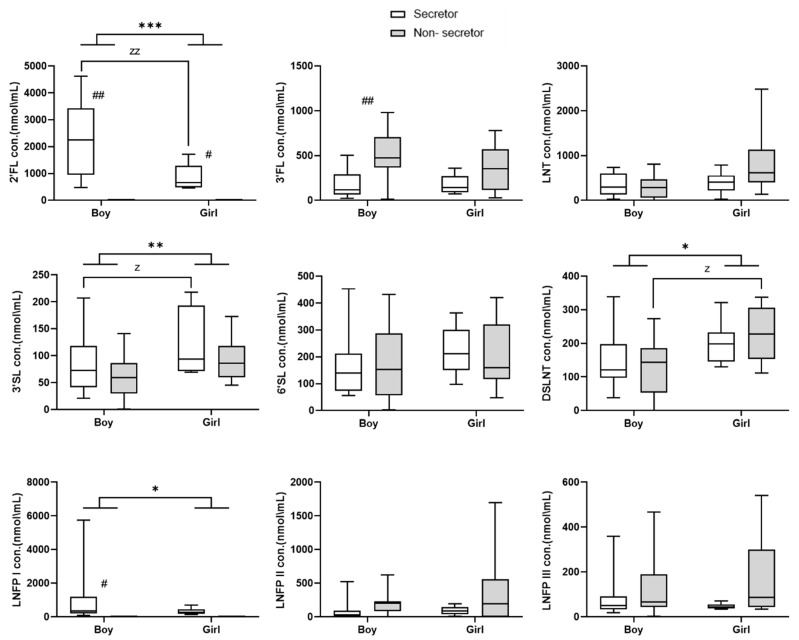
Box plot showing concentrations (mean ± SEM) of the individual HMOs by secretor status and infant sex. Boy: secretors, *n* = 20 (8 mothers); non-secretors, *n* = 9 (3 mothers). Girl: secretors, *n* = 9 (3 mothers); non-secretors, *n* = 15 (6 mothers). # *p* < 0.05 and ## *p* < 0.01 indicate differences between secretors and non-secretors by infant sex. z *p* < 0.001 and zz *p* < 0.0001 indicate differences within the group (secretors or non-secretors) between the sexes. * *p* < 0.05, ** *p* < 0.01, *** *p* < 0.001 indicate differences in infant sex without considering the secretor status. Multivariant analysis was used for comparisons, and detailed results are presented in Table 2. SEM = standard error of mean.

**Table 1 nutrients-15-02548-t001:** Maternal and neonatal characteristics of the cohort (N = 20).

Maternal age, year	32.9 ± 0.9 (25–43)
Non-secretor	9 (45)
Number of HM samples	23 (44.2)
Secretor	11 (55)
Number of HM samples	29 (55.7)
Lewis negative	3 (15)
Lewis positive	17(85)
Milk group phenotype	
*Se+Le+*	10 (50)
*Se−Le+*	7 (35)
*Se+Le−*	1 (5)
*Se−Le−*	2 (10)
Maternal Pre-pregnancy BMI [kg/m^−2^]	22.4 ± 0.6 (16.7–26.2)
Non-smoker	18 (90)
Smoker before pregnancy	2 (10)
Omnivorous diet	17 (85)
Vegetarian diet	3 (15)
Infant gestational age, week	38.15 ± 0.5 (33–41)
Vaginal delivery	17 (85)
C-section delivery	3 (15)
Infant birth weight, g	3084 ± 149.7 (1900–4056)
Male infant	11 (55)
Female infant	9 (45)
Not admitted to the NICU	17 (85)
Admitted to the NICU	3 (15)
Parity	
1	9 (45)
2	7 (35)
3	4 (20)
Season of collected samples (n = 52)	
Summer (June–August)	44 (84.6)
Other (September–May)	8 (15.4)

Data are expressed as mean ± SEM (range) or n (%).

**Table 2 nutrients-15-02548-t002:** Multivariable analysis results of HMO’s.

2′-FL	Variables	B-Coeff	SEM	*p*
	Time 1 vs. 0	−749.320	191.250	<0.001
	Time 2 vs. 0	−838.593	205.317	<0.001
	Time 2 vs. 1	−89.273	205.613	0.664
	GA	177.563	76.041	0.02
	Sex	−1359.295	361.798	<0.001
	Diet	957.156	514.795	0.063
3-FL	Variables	B-Coeff	SEM	*p*
	Time 1 vs. 0	38.032	50.656	0.453
	Time 2 vs. 0	4.163	54.122	0.939
	Time 2 vs. 1	−33.869	54.229	0.532
	Age	−6.531	5.692	0.251
	ppBMI	21.437	10.317	0.038
	GA	−24.585	11.318	0.03
	FUT2	−264.571	53.332	<0.001
	FUT3	374.080	83.835	<0.001
	MOD	−0.614	66.082	0.993
	Season	71.280	69.251	0.303
	Diet	39.044	93.166	0.675
3′-SL	Variables	B-Coeff	SEM	*p*
	Time 1 vs. 0	69.642	8.953	<0.001
	Time 2 vs. 0	−83.186	9.501	<0.001
	Time 2 vs. 1	−13.544	9.539	0.156
	Sex	28.204	10.523	0.007
	FUT2	30.478	10.113	0.003
	Smoking	−11.493	16.783	0.493
6′-SL	Variables	B-Coeff	SEM	*p*
	Time 1 vs. 0	60.097	26.095	0.021
	Time 2 vs. 0	−59.359	27.654	0.032
	Time 2 vs. 1	−119.455	28.018	<0.001
	ppBMI	15.024	5.151	0.004
	MOD	73.711	34.826	0.034
	NICU	133.848	38.868	0.001
DSLNT	Variables	B-Coeff	SEM	*p*
	Time 1 vs. 0	8.762	19.167	0.648
	Time 2 vs. 0	−85.223	20.223	<0.001
	Time 2 vs. 1	−76.461	20.430	<0.001
	Sex	49.538	22.483	0.028
	FUT3	−61.026	28.822	0.034
	MOD	47.709	30.700	0.12
LNFP I	Variables	B-Coeff	SEM	*p*
	Time 1 vs. 0	−43.467	180.402	0.810
	Time 2 vs. 0	−166.577	193.303	0.389
	Time 2 vs. 1	−120.275	194.590	0.537
	Sex	−711.901	343.940	0.038
	FUT3	−327.260	599.268	0.585
	Diet	1526.702	599.044	0.011
LNFP II	Variables	B-Coeff	SEM	*p*
	Time 1 vs. 0	−114.548	32.480	<0.001
	Time 2 vs. 0	45.990	34.820	0.187
	Time 2 vs. 1	−68.558	34.869	0.049
	FUT2	−126.062	104.624	0.228
	MOD	407.329	140.352	0.004
	Season	410.123	130.796	0.002
LNFP III	Variables	B-Coeff	SEM	*p*
	Time 1 vs. 0	−47.673	23.150	0.039
	Time 2 vs. 0	−74.265	24.571	0.003
	Time 2 vs. 1	−26.592	24.736	0.282
	GA	−24.039	6.206	<0.001
	FUT2	−23.760	28.448	0.404
	Season	138.329	36.950	<0.001
LNT	Variables	B-Coeff	SEM	*p*
	Time 1 vs. 0	−265.735	72.942	<0.001
	Time 2 vs. 0	97.095	77.924	0.213
	Time 2 vs. 1	−168.640	78.106	0.031
	Sex	233.535	161.251	0.148
	MOD	396.811	225.609	0.079
	Season	382.223	183.644	0.037

Abbreviations: B-Coeff, B-coefficient; SEM, standard error mean; GA, gestational age; MOD, mode of delivery; ppBMI, pre-pregnancy body mass index.

## Data Availability

The data presented in this study are available on request from the corresponding author. The data are not publicly available due to ethical considerations.

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
