# Peer review of "Human Milk Oligosaccharide Profile across Lactation Stages in Israeli Women—A Prospective Observational Study"

_nutrients, 2023, doi:10.3390/nu15112548_

Round 1
Reviewer 1 Report
In this study, the authors determined the concentrations of the representing human milk oligosaccharides (HMOs) in colostrum, transitional milk and mature milk in Israeli women. Although many paper have been published for the concentrations of HMOs in breast milk in some area in the world, this is the first case for Israeli women. As I acknowledge the significance for the collection of such data in many areas in the world, I basically agree the acceptance for the publication of the paper. However., I have a few questions. They should answer for the followings.
1. The determination of the concentrations of HMOs was performed by HPLC with HILIC column and mass spectrometry at first. HMOs were not reduced with NaBH4 prior to the HPLC. Did not they observe two peaks by α and β anomers of reducing ends for each HMO in the HPLC profile? In the HPLC with graphite carbon column, two peaks for each HMO are seen. I suggest that the representing HPLC profile with the annotation of each peak will be shown in a figure.
2. At second, the determination of the concentrations was performed by gas chromatography – mass spectrometry. I assume that two peaks should be seen for each HMO, because they are not reduced. I suggest that the representing GC profile with the annotation of each peak will be shown in a figure, too. Were not some peaks of HMOs overlapped with other ones?
3. Was the concentration value of each HMO described in the figures of the paper obtained by the determination with HPLC-MS or GC-MS? They should clearly describe.
Other comments for several points are follows.
1) Abstract line 25
Mass spectrometry chromatograms should be corrected as mass spectra.
2) Introduction line38~line 67
I suggest that they will add the basic information of HMOs as followings.
Human breast milk contains 12 ~ 13 g/L of HMOs, of which more than 250 varieties have been separated and more than 200 chemical structures have been characterized to date.
Ashline, D.J. et al. (2014) Structural characterization by multistage mass spectrometry (MSn) of human milk glycans recognized by human rotaviruses. Mol. Cell. Proteomics, 13, 2961-2974.
Remozora, C.A. et al. (2018) Creating a mass spectral reference library for oligosaccharides in human milk. Anal. Chem. 90, 8977-8988.
Of 200 characterized HMOs, around 170 structures are listed in the tables.
Urashima, T. et al (2018) Human milk oligosaccharides as essential tools for basic and application studies on galectins. Trends Glycosci. Glycotechnol. 30, SE51-SE65.
Urashima, T., Katayama, T., Fukuda, K., Hirabayashi, J. Human milk oligosaccharides and innate immunity in Comprehensive Glycoscience, the second edition (Barchi, J. ed.), Amsterdam, Elsevier, 2021. Doi: 10.1016/B978-0-12-819475-1.00009-2
The concentrations of the representing HMOs have been determined in Secretor and non-Secretor donor’s milk at several lactation periods.
Thurl, S. et al. (2017) Systematic review of the concentrations of oligosaccharides in human milk. Nutr. Rev., 75, 920-923.
The average values of the concentrations of 2’-FL, 3-FL, LNT, 3’-SL and 6’-SL have been calculated from the published data by weighted analysis.
Conza, D.B. et al. (2022) Weighted analysis of 2’-fucosyllactose, 3-fucosyllactose, lacto-N-teeraose, 3’-sialyllactose, and 6’-sialyllactose concentrations in human milk. Food and Chemical Toxicology 163, 112877.
3) Introduction line38~line 67
I suggest that some references of the determinations of HMOs in other area in the world.
China
Austin, S. et al. Temporal change of the content of 10 oligosaccharides in the milk of Chinese Urban mothers. Nutrients 2016, 8(6), 348
Mexico
Chaturvedi, P. et al. Fucosylated human milk oligosaccharides vary between individuals and over the course of lactation. Glycobiology 2001, 11, 365-372
Vietonam
Van Leeuwen et al. Regional variations in human milk oligosaccharides in Vietonam suggest FucTx activity besides FucT2 and FucT3. Sci. Rep. 2018, 8, 16790
Samoa
Leo, F. et al. etermination of sialyl and neutral oligosaccharide levels in transition and mature milks of Samoan women, using anthranilic derivatization followed by reverse phase high performance liquid chromatography. Biosci. Biotech. Biochem. 2010. 74(2), 298-303.
4) Line 85, 86
“presence” should be “presence/absence”.
5) Line 89 and others
“3’-fucosyllactose (3’-FL)” must be “3-fucosyllactose (3-FL)”.
6) Line 170
“theirs” must be “their”.
7) Line174 ~ line 176
Are the data of the concentrations for only Secretor donors or average values for both Secretors and non-Secretors? They should describe it.
8) Discussion
I suggest that they compare the concentration values of HMOs with the previous data for the breast milk in other area in the world. Are the values for HMOs similar to other ones?
9) Line 209
The citation here may not be [25] (Thurl, S. Nutr. Rev., 2017, 75, 920-933 but the following.
Zinger-Yosovich, K.D. et al. Blocking of Pseudomonas aeruginosa and Chromobacterium violaceum lectins by diverse mammalisn milks. J. Dairy Sci. 2010, 93, 473-482.
10) Line 215
The authors of ref 28 are not J, B,; RH, B. et al. but Bienenstock, J. et al. This mist be corrected.
11) Line 231 after disease NEC.
I suggest that the following may be added.
This was shown by in vivo experiments with neonatal rat model.
Jantscher-Krenn, E. et al. The human milk oligosaccharide disialyllacto-N-tetraose prevents necrotizing enterocolitis in neonatal rats. Gut 2012, 61, 1417-1425.
12) Line 232
The followings may be cited here.
Autran, C.A. et al. Human milk oligosaccharide composition predicts risk of necrotizing enterocolitis in preterm infants. Gut. 2018, 67, 1064-1070.
Masi, AS. Et al. Human milk oligosaccharide DSLNT and gut microbiome in preterm infants predicts necrotizing enterocolitis. Gut, 2021, 70, 2273-2282.
13) Line 255
Most HMO’s content other than 3-FL
14) Line 265 after neurons
The following may be inserted here.
Feeding 3’-DL and 6’-SL increased the level of gangliosides in cerebellum and corpus callosum in in vivo experiment with piglets.
Jacobi, S.K. et al. Dietary isomers of sialyllactose increase gangliodide sialic acid concentrations in the corpus callosum ans verebellum and modulate the colonic microbiota of formula-fed piglets. J. Nutr. 2016, 146, 200-208.
Author Response
The point-by-point response to the reviewer’s comments are uploaded in the attached file

Reviewer 2 Report
The manuscript evaluated a special amino acids-based infant formula supplemented with two HMOs on safety, growth, tolerability, and gut microbiome for infants with CMPA. Overall, the manuscript is well written, however, some information is missing which lowers the scientific value.
1. The nutrients composition of this specific infant formula is not shown, and makes the research less reproducible.
2. Please indicate why the concentrations, i.e. 1.0g/L and 0.5g/L of the two HMOs were chosen.
3. Please indicate the origins of the two HMOs.
4. The nutrients needed for infants vary among different ages, as well as the gut microbiota development, just as the results shown in Figure 6 of this study, so the infants age 1-8 months seems too wide.
5. It is not realistic to include negative controls for infants study, however, healthy positive controls should be involved.
6. How the number of the infants were determined?
7. SCFAs need to be absorbed to maximize their healthy benefits, how to explain the higher SCFAs concentration in the stool sample can demonstrate a positive result?
Author Response
The point-by-point response to the reviewer’s comments are uploaded in the attached file.

Reviewer 3 Report
Dear authors, thank you for a short concise paper on breast milk HMO composition.
Minor revisions: However, this reviewer have important corrections to the analytical section line 87-118 before recommending publication of the manuscript:
1. Line 98: Purchasing of standards for HPLC, LC-MS and GC-MS from which vendor? Should be placed in the methods section under materials.
2. How was the error-bars indicated on Fig. 1 determined. Was the analytical error determined - single, double or triplicate determinations.
3. By which method was the quantification in nmol/mL (Fig. 1) performed for each of the HMO's? The authors indicate GC-MS, LC-MS, HPLC. It is suggested a table with each HMO and analytical method + detector used for quantification + m/z ion or transition used for the quantification. IF PDA detector is used please indicate the UV signal (sugars have no UV signal??). If LC-MS method is used then the SRM/MRM conditions mother ion/fragment m/z used for the transition is needed to complete the analytical information. If GC-MS is used then the derivative + m/z used for the quantification should be used.
4. If the same HMO/analyte was measured by more than one method please indicate which was used or was the average used as the result.
5. If methods for the literature has been adapted for the analytical procedures, please insert references to these as well as the above.
Author Response

(The authors gave the same response as above.)

Reviewer 4 Report
This is an interesting work that provides longitudinal follow-up of HMOs in breast milk in an Israeli population and therefore deserves to be published.
There are two aspects that I would like to comment on:
1.-On line 55 it says HBM and I don't know if it should say HM.
2.-I do not see the need for figures 3 and 4 since the seasonal differences and the differences by sexes are very clear in table 2.
Author Response

(The authors gave the same response as above.)
